# Injuries Associated with the Practice of Calm Water Kayaking in the Canoeing Modality

**DOI:** 10.3390/jcm10050902

**Published:** 2021-02-25

**Authors:** Manuel Isorna-Folgar, Raquel Leirós-Rodríguez, Rubén Paz-Dobarro, Jose Luis García-Soidán

**Affiliations:** 1Faculty of Education and Social Work, Universidade de Vigo, Edificio de Ferro, Campus As Lagoas s/n, 32004 Ourense, Spain; isorna.catoira@uvigo.es; 2SALBIS Research Group, Department of Nursing and Physiotherapy, Faculty of Health Sciences, Universidad de Leon, Campus of Ponferrada s/n, 24401 Ponferrada, Leon, Spain; 3Faculty of Physical Therapy, Universidade da Coruña, Campus de Oza s/n, 15006 A Coruña, Spain; ruben_pazdobarro@hotmail.com; 4Faculty of Education and Sport Sciences, Universidade de Vigo, Campus a Xunqueira, s/n, 36005 Pontevedra, Spain; jlsoidan@uvigo.es

**Keywords:** assessment, injury and prevention, competition, gender, overtraining

## Abstract

Knowing the most frequent injuries in canoeists is important, considering the consequences for the athlete’s sports career, health, and labour, social and economic life. Therefore, the aim of this study was to describe the most frequent injuries among high-level canoeists and the intrinsic variables of the sport’s practice (years of practice, number of training sessions per week, and stretching habit) that can influence the appearance of such injuries. An observational, transversal, and retrospective epidemiological study was carried out with 122 canoeists that completed a sport injuries questionnaire (number, body area, type, and severity of injuries). The ratio of injuries per participant was 1.1 injuries/year in men; and 1.5 injuries/year in women. Shoulder injuries were the most frequent, followed by knee injuries (in women) and lower back (in men), and the other segments of the upper limbs. In men, injuries occurred more frequently in the central period of training sessions and in women during the last 15 min of training sessions. Then, there is a common profile of injuries in canoeists: Being a female, having more years of sports practice, and never executing stretching exercises are associated with predisposing factors to injuries.

## 1. Introduction

Calm water kayaking is a sport modality that entered the Olympic Games in the year 1936, and which is performed in distances of 500 and 200 m for women and 1000 and 200 metres for men, the kayaking and Canadian canoeing disciplines [1,2]. Although kayaking is one of the sports in which Spain reached the podium on most occasions, it is still unknown to many people. Currently, in Spain, according to the Royal Spanish Canoe Federation, there are around 12,000 active federated canoeists belonging to 258 registered clubs, and the number of people who practise this sport is growing. However, the resources allocated to research on this particular sport are very scarce [3]. Kayaking has numerous modalities, with the most important ones being water calm kayaking and Slalom, which are within the Olympic programme. In water calm, there are two modalities: kayaking and canoeing. The latter was the object of the present study and it differs from the former for being unilateral and asymmetric, where the paddling is performed at one side with a single-bladed paddle oar, with the paddler leaning on the boat on his/her opposite knee and foot [4,5].

This is a water sport in which movements are repeated and force is applied in a repeated manner, mainly using the trunk and upper limbs, against the resistance force exerted by the water on the vessel (canoe). There are two well-differentiated disciplines within calm water kayaking based on the position of the boat and the paddling movement: the kayaker performs the paddling at both sides of the boat in a sitting position with his/her lumbar rachis flexed, whereas the canoeist paddles at one side only, performing a cyclic movement of flexion, rotation and lateral tilt [6]. These differences in the paddling technique have been associated with certain specific adaptations in rachidial morphotype and pelvic arrangement [7]. To understand these adaptations and the injury risk factors of this discipline, it is necessary to analyse in detail the sporting gesture of paddling in a canoe. This, for its biomechanical analysis, can be divided into three phases: (a) entry of the paddle into the water or attack; (b) application of maximum force in the middle part or traction; and (c) exit of the shovel from the water or extraction and recovery [2,5]. During the attack, the paddle is in the most distant position from the canoeist because he is with the lower arm extended and the upper arm semi-extended above the head. The trunk meets with an inclination of 45° and rotation of 25°. Lastly, the front knee meets a 90° flexion. The shovel begins to submerge, lowering the arms, while the trunk increases its anterior flexion. During the traction phase, the paddle is vertical and in the water, because the canoeist is with his lower arm extended and close to the surface of the water. The arm in the upper position maintains the position above the head. The trunk begins to rise with the end of the twist. Subsequently, the blade begins to lose perpendicularity while the trunk extends and rotates towards the shovel side. Meanwhile, the pelvis is posteriorized (retroversion movement) and the hips move posteriorly until they align. During extraction and retrieval, the paddle comes out of the water, with the canoeist’s trunk upright and the shoulders facing forward, the pelvis slightly forward (small-magnitude anteversion movement) and rotating to the side of the paddle

There is an increasing number of canoeists who suffer from injuries, partly due to the psychophysical demands to which these athletes are subjected when they reach a high-performance level [8], as well as to the greater pressure from the media on professional athletes, constantly requesting them to win. The latter cause also reaches non-professionals, leading them to adopt behaviours that may become dangerous for their physical integrity [9].

Describing the aetiology of injuries in canoeing, knowing the factors that influence the rate of injuries, and the need for understanding why these injuries occur have been some of the main drivers of different studies [10,11,12]. Knowing the most frequent injuries in canoeists is important, considering the consequences for the athlete’s sports career [13], health [14], and labour, social and economic life [15]. Thus, detailed information about the rate of morbidity of the different injuries and the early identification of the associated epidemiological factors will allow the implementation of prevention programmes and intervention strategies, which would help to reduce the consequences of the most frequent injuries among canoeists. Therefore, the aim of the present study was to describe the most frequent injuries among high-level canoeists and the intrinsic variables of the sport’s practice (years of practice, number of training sessions per week and stretching habit) that can influence the appearance of such injuries.

## 2. Materials and Methods

This was an observational, transversal and retrospective epidemiological study. It was conducted during the Spanish Calm Water Kayaking Championship. Considering the total of 158 canoeists who participated in all the categories of this championship, a necessary sample size of 119 cases was established to reach the 97% confidence level, 50% of heterogeneity and 5% margin of error. Taking these data into account, reaching this sample size ensures that the variability that implies not analysing all of the championship participants does not significantly modify the results obtained.

The inclusion criteria to participate in this study were: (a) being over 16 years of age; (b) participating in the canoeing modality; (c) being federated in the Royal Spanish Kayaking Federation; (d) having practised canoeing for more than 2 years; (e) having the minimum mark to participate in the Spanish championship in the junior or senior category; (f) submitting a written informed consent and, if under 18, a parental authorisation. Finally, a total of 122 canoeists participated in the study: 99 men (81.1%) and 23 women (18.9%), with an average age of 23.3 ± 8.2 years.

For the sample selection, firstly, the coaches of the clubs were contacted after the publication of the registrations of the participants in the 2019 Spanish Calm Water Kayaking Championship in the canoeing modality, and the conditions of the study were explained to them. The purpose of the study was explained to them again the day before the beginning of the championship, specifying the data gathering method. All participants were requested to submit a written informed consent and, for the underage participants, an authorisation from their parents or legal guardians, guaranteeing the confidentiality and anonymity of the data. The study respected the ethical principles of the Declaration of Helsinki (2008) and Law 15/1999 on Data Protection. All questionnaires were administered individually in the presence of one of the researchers. This research obtained ethical approval from the Commission of Ethics of the University of A Coruña (Spain) (code: 2-0406-14).

For the gathering of data about injuries, we used the self-report models about sports injuries described by Olmedilla et al. [16] and the questionnaire developed by Díaz et al. [17]. Both methods were used to evaluate the history of injuries of the participants and to gather information about their sport and personal data. For the gathering and classification of the injuries, the following criteria were taken into account: their number and severity, type, area and time of occurrence. This provided information, in a retrospective manner, about the number, body area, type and severity of the injuries suffered during the previous seasons, as well as other affiliation data. In order to verify that the diagnosis of the injury was correct, an additional item showed which healthcare professional had performed the diagnosis and whether medical tests had been performed on the participant.

The severity of the injuries was established based on the following classification [18]: (a) mild: required treatment or not and interrupted, at least, one day of training; (b) moderate: required treatment and forced the canoeist to interrupt six days of training, and even some competition; (c) severe: resulted in one to three months of sports injury leave and required treatment; (d) very severe: resulted in four or more months of sports injury leave and may even have required hospitalisation, surgical intervention and rehabilitation.

To determine the descriptive data, the following statistics were used: mean, standard deviation percentages and frequencies. To establish the differences between sexes, the Student’s *t*-test was conducted. To compare the number of injuries with the quantitative variables of training (years of sport practice, number and length of the weekly sessions) and with the extrinsic qualitative variables of training, the ANOVA test was performed, using Bonferroni’s test.

The significance level was established at *p* ≤ 0.05. The statistical software SPSS v21.0 for Windows (IBM©) was used for all statistical analyses.

## 3. Results

The average age of the sample was 23.3 ± 8.2 years (men: 25.7 ± 9.2 years; women: 21.3 ± 3.9 years), with the group of women being significantly younger (*t*-test *p* < 0.01).

Table 1 shows that the males had significantly more years of sport experience than the females (x^2^ = 45.5; *p* = 0.03). On the other hand, regarding the frequency of training sessions, both sexes showed a similar pattern (*p* > 0.05).

With respect to injuries, the females showed a significantly greater rate compared to the males in the period of 12 months prior to the championship (91.3% vs. 60.6%; *t*-test *p* < 0.001) and, of all the recorded injuries (*n* = 135), 25.9% were suffered by the female canoeists (although they represented only 18.9% of the sample). The ratio of injuries per participant was 1.1 injuries/years (1.5 injuries/year for the females). On the other hand, the severity of the injuries was not significantly different between sexes (Table 1).

Lastly, the type of injuries suffered was not significantly between sexes, with muscular injuries being the most prevalent for both males and females, followed by tendinous, ligamentous and bone injuries (Table 1).

Regarding the distribution of the injuries in the different anatomical segments, the most frequent ones were shoulder injuries, followed by knee injuries (in females), lumbar injuries (in men) and injuries in the rest of the upper limb segments (Figure 1). The result of the chi-square test was not statistically significant for the anatomical distribution of the injuries according to gender. On the other hand, with respect to the time of occurrence of the reported injuries, the rate of injuries was significantly greater during training sessions than during competitions (x^2^ = 44.3; *p* < 0.001) for the entire sample (males: x^2^ = 27.3; *p* < 0.001) (females: x^2^ = 29.4; *p* < 0.001).

Figure 2 indicates that, in the males, the injuries occurred more frequently in the central period of the training sessions, whereas, for the females, the greatest frequency corresponded to the last 15 min of the training sessions.

Regarding the habit of stretching, it was found that, of the total of 122 analysed cases, 102 performed stretching exercises, and 18 performed no stretching exercises at all. The proportion of stretching habits was divided as follows: 14.8% of the participants performed no stretching, 54.1% stretched after the training session, 28.7% performed stretching exercises before and after the training sessions, and 2.5% only stretched before training. The prevalence of the habit of stretching and the moment in which they decided to do it was not different between the sexes (*p* > 0.05). The rate of injuries was significantly greater in the canoeists who performed no stretching exercises (x^2^ = 36.1; *p* < 0.001) for the entire sample (males: x^2^ = 23.4; *p* < 0.001) (females: x^2^ = 32.6; *p* < 0.001).

The bivariate analysis showed a positive correlation between the number of practice years and the injury rate (r = 0.7; *p* < 0.01). However, the number of training sessions per week was not correlated with the injury rate (*p* > 0.05).

## 4. Discussion

The aim of this study was to describe the most frequent injuries that occur in high-level canoeists and the intrinsic variables of the sport practice (years of practice, number of training sessions per week and stretching habit) that can influence the appearance of such injuries. In view of the obtained results, it can be confirmed that the injury rate in one year was 66.4% among the analysed canoeists, with an average occurrence of one injury per year, and that this phenomenon is related to certain extrinsic variables such as the years of sports practice and the habitual execution of stretching exercises.

With respect to the main variable of this study (injury area), and considering the characteristics of the canoeing modality, where the paddling technique is cyclic (using mostly the musculature of the upper limb), there was a greater number of injuries in the shoulder area for both sexes compared to the rest of the anatomical segments. This phenomenon could be due to the fact that this is a key area in the technical performance of this sport, an excessive use, post-traumatic stress (since the potential energy applied to each paddling stroke is transmitted directly to this joint), or even to the impact of the paddle on the bottom in shallow places. In addition, a great similarity was found in the anatomical regions with the study of Treus et al. [19], highlighting that, in addition to coinciding in this predominant injury area, the second area with greater injury rate, i.e., the lumbar segment (with 10.7%), was similar in both studies. Moreover, considering that the most frequent time when the injuries appeared was during the second half of the training sessions, it can be asserted that consistent overtraining can logically lead to muscular imbalances, glenohumeral and scapular kinematic dysfunctions, soft tissue damage and inflammatory processes. If the athlete wants to reach the maximum sports performance, prevention programmes are required [20], which should be specifically focused on correcting asymmetries and muscular strength imbalances between agonist and antagonist muscles [21].

Competition usually leads to a pattern of injuries of the soft tissues that affect the upper limb, with the shoulder area being reported most frequently in different studies, which can cause permanent damage [20,22]. Abraham and Stepkovitch [4], in a different calm water modality, corroborated these findings, observing shoulder injuries as the most frequent, in this case with 35.6%, followed by spinal injuries (23%) and lumbar injuries (17%). Lastly, regarding anatomical localisation, other authors highlighted that the greatest percentage of injuries in kayaking also occurred in the shoulder [23,24]. On the other hand, Bell et al. [25] concluded that professional rowing is relatively less harmful than other water sports, presenting a general risk of 1.8 injuries per 1000 h of exposure, and that most of the injuries are not severe, although a significant percentage of them ranged from moderate to severe. These researchers found that most of the injuries that occur in canoeing are shoulder sprains and distensions, followed by spinal and arm or wrist injuries and that a greater percentage of injuries appear in long-distance canoeists among the reported injuries [25,26].

Regarding the injury rate in males and females, this was greater in the latter. However, for the modality of the marathon in kayaking, Abraham and Stepkovitch [4] not only asserted that women are equally prepared to suffer from injuries as men are, but that men are 3.6 times more likely to suffer from an injury than women. In their study, they also associated this finding with the fact that men train more often and have been practising this modality for a longer time; such correlation has also been detected in the present study.

The protective effect of stretching against injuries identified in the results is in line with the scientific evidence available. Flexibility decrease and muscular weakness can cause injuries, and stretching exercises and the improvement of flexibility are effective in the prevention of injuries [27]. Moreover, consistent stretching improves not only flexibility but also the resistance of muscular and tendinous tissues against mechanic stress [28], and these factors are strongly associated with the types of injuries reported by the participants.

The present study has some limitations that must be pointed out: (a) regarding the specific perception of pain of each athlete, the inherent subjectivity and individuality of pain tolerance may lead to an injury being undetected or exaggerated; thus, this problem would alter the results of the analysis of this study; (b) with respect to the participants with very severe injuries, as is shown by the statistical data, this percentage of the sample was the lowest, although this could be due to the fact that most athletes with a severe injury have abandoned this sport and, therefore, did not complete the questionnaire; (c) the population of females in this study was small, due to the recent establishment of this category, and, although the sample was representative, including almost all the people who practise this modality in Spain, all of whom met the inclusion criteria, this population is still significantly smaller than the male population; (d) although the pattern of injuries analysed corresponds to what should be the gold standard of technique (given that they are the canoeists with the best results in Spain), the results obtained may not be extrapolated to the total number of recreational or semi-professional canoeists. It must be taken into account that, considering the sample universe of 12,000 federated canoeists in Spain, the statistical representativeness of the data obtained is reduced to 50% heterogeneity, 8% margin of error and 92% confidence level; and (e) the specific type of injuries or their origin in direct trauma was not taken into account, a variable that should be analysed in future research. For all the above, new investigations should be carried out on the prevalence, incidence, and etiopathogenesis of musculoskeletal injuries in canoeists that analyse the evolution of the injuries, their possible recurrence and the rate of abandonment due to serious injuries and its relationship with sports technique and posture sway during canoeing [29,30].

## 5. Conclusions

It can be confirmed that there is a common profile of injuries in canoeists, with the shoulder being the anatomical segment with the largest number of injuries. The severity of the most frequent injuries is low, interrupting at least one day of training. Being a female, having more years of sport practice and never executing stretching exercises are associated with predisposing factors to injuries. Consequently, all these factors must be taken into account to prevent them and must be considered when designing training programs for these athletes.

## Figures and Tables

**Figure 1 jcm-10-00902-f001:**
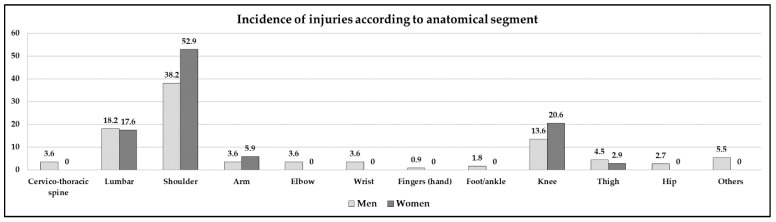
Incidence of injuries according to the anatomical segment.

**Figure 2 jcm-10-00902-f002:**
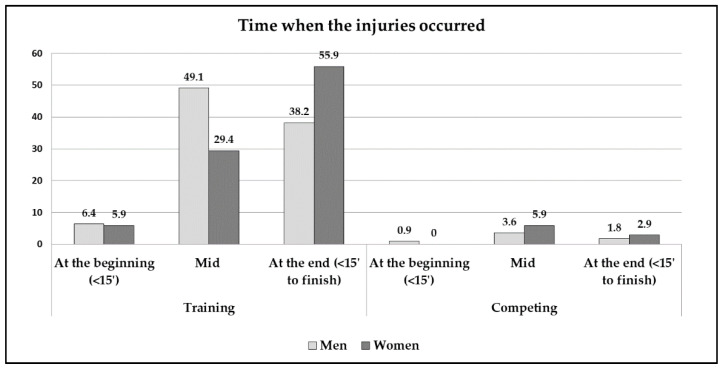
The time when the injuries occurred.

**Table 1 jcm-10-00902-t001:** Descriptive analysis of the sample.

	All (*n* = 122)	Men (*n* = 99)	Women (*n* = 23)
Years of practice			
Less than 2	10 (8.2%)	5 (5.1%)	4 (17.4%)
Between 2 and 5	29 (23%)	19 (19.2%)	5 (21.7%)
Between 5 and 10	41 (32.8%)	31 (31.3%)	9 (39.2%)
More than 10	88 (36.1%)	44 (44.4%)	5 (21.7%)
Number of training sessions			
Less than 3	7 (5.7%)	4 (4%)	3 (13.1%)
Between 3 and 5	33 (27.1%)	28 (28.3%)	5 (21.7%)
More than 5	82 (67.2)	67 (67.7%)	15 (65.2%)
Injuries in the past year			
Yes	81 (66.4%)	60 (60.6%)	21 (91.3%)
No	41 (33.6%)	39 (39.4%)	2 (8.7%)
Injury severity			
Mild	65 (48.1%)	49 (49%)	16 (45.7%)
Moderate	51 (37.8%)	38 (38%)	13 (37.1%)
Severe	16 (11.9%)	11 (11%)	5 (14.3%)
Very severe	3 (2.2%)	2 (2%)	1 (2.9%)
All	135 (100%)	100 (74.1%)	35 (25.9%)
Injury type			
Muscle	96 (71.1%)	72 (72%)	24 (68.6%)
Bone	3 (2.2%)	2 (2%)	1 (2.9%)
Tendinous	29 (21.5%)	21 (21%)	8 (22.8%)
Ligamentous	5 (3.7%)	3 (3%)	2 (5.7%)
Contusion	1 (0.7%)	1 (1%)	0
Others	1 (0.7%)	1 (1%)	0
All	135 (100%)	100 (100%)	35 (100%)

## Data Availability

The data presented in this study are available on request from the corresponding author.

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
