# Peer review of "Injuries Associated with the Practice of Calm Water Kayaking in the Canoeing Modality"

_jcm, 2021, doi:10.3390/jcm10050902_

Round 1

Reviewer 1 Report

I'm very pleased to do a review of this manuscript. The paper is well written. Congratulations to the Authors. I have only few comments. 

Introduction

Generally, the introduction is fine, however it might be too short. I also prefer fewer paragraphs, preferably 3-4 , where the third describes the rationale of the study and the fourth - the aim of the study.

Materials and Methods

I am a little worry about about small percentage of female population in the study but the Authors explained it at the end of Discussion - as one of the limitation of their study. The same concerns specific perception of pain of each athlete, what is also explained. 

Results

Very well written part, with apropriate table and figures.

Discussion

Interesting discussion, with the limitations of the study clearly described.

Conclusion

It is also well written, but it might be stronger.

Author Response

Dear Editor and Reviewer of Journal of Clinical Medicine:

Thank you very much for your suggestions and contributions to improve the quality of the manuscript. Following your indications, we respond, point by point, to the reviewers' comments.

In the text, all the modified or added sentences have been written in red to facilitate the correction by the reviewers.

  1. Introduction: generally, the introduction is fine, however it might be too short. I also prefer fewer paragraphs, preferably 3-4 , where the third describes the rationale of the study and the fourth - the aim of the study.

The authors have expanded the Introduction and reorganized the paragraphing of this section.

  1. Materials and Methods: I am a little worry about about small percentage of female population in the study but the Authors explained it at the end of Discussion - as one of the limitation of their study. The same concerns specific perception of pain of each athlete, what is also explained.

Thank you very much for taking into account the efforts that the authors have made to be honest with the limitations of this research.

  1. Results: Very well written part, with apropriate table and figures.

Discussion: Interesting discussion, with the limitations of the study clearly described.

Conclusion: It is also well written, but it might be stronger.

The authors greatly appreciate your positive assessment.

Once again, thank you very much for the time spent and the interest shown in this work; as well as in the positive evaluations you have given of it.

Receive a warm greeting,

The authors.

Reviewer 2 Report

This is a cross-sectional study to show the injury rate in sport activities.  It is worth to show the pattern of injuries of canoeists. However, there are some methodological problems to conclude the results.

1. In this study, the representativeness of samples is critical to the study result. Authors describes that there are 12,000 active federated canoeists and 158 canoeists who participated in the championship were selected as subjects.  And 122 were participated in the study. 1) Do authors believe that the injury pattern of 158 are same as that of the 12,000 canoeists? If yes, please describe or prove it in the manuscript. 2) Do authors believe that the injury pattern of 122 are same as that of 158 canoeists? If yes, please describe or prove it in the manuscript. Why were 36 dropped. 

2. The inclusion criteria were over 16 years age and more than 2 years of experience. It means that authors included subjects who started canoeing at age of 14.  Is there any criteria to select subjects at age of 14?

3. Even though the abandoned canoeists who had a severe injury was described in the limitation, it is critical to give more information of severe injured who eventually quit canoeing to describe injury pattern of canoeing. 

4.  Information on stretching between men and women should be provided because stretching and sex are important factors to have injuries. 

Author Response

Dear Editor and Reviewer of Journal of Clinical Medicine:

Thank you very much for your suggestions and contributions to improve the quality of the manuscript. Following your indications, we respond, point by point, to the reviewers' comments.

In the text, all the modified or added sentences have been written in red to facilitate the correction by the reviewers.

  1. In this study, the representativeness of samples is critical to the study result. Authors describes that there are 12,000 active federated canoeists and 158 canoeists who participated in the championship were selected as subjects. And 122 were participated in the study. 1) Do authors believe that the injury pattern of 158 are same as that of the 12,000 canoeists? If yes, please describe or prove it in the manuscript.

We have added this detail as a limitation of the investigation (at the end of the Discussion).

  1. Do authors believe that the injury pattern of 122 are same as that of 158 canoeists? If yes, please describe or prove it in the manuscript. Why were 36 dropped.

The analysis of 122 canoeists exceeds the necessary limit of 119 participants to be able to reach the 97% confidence level, 50% of heterogeneity and 5% margin of error.

This sample size calculation is detailed and justified at the beginning of Material and Methods.

  1. The inclusion criteria were over 16 years age and more than 2 years of experience. It means that authors included subjects who started canoeing at age of 14. Is there any criteria to select subjects at age of 14?

Subjects under 16 years of age could not participate since they are not allowed to participate in the Spanish Calm Water Kayaking Championship.

  1. Even though the abandoned canoeists who had a severe injury was described in the limitation, it is critical to give more information of severe injured who eventually quit canoeing to describe injury pattern of canoeing.

We've added that detail after research limitations (at the end of the Discussion section).

  1. Information on stretching between men and women should be provided because stretching and sex are important factors to have injuries.

The authors have analyzed by sex the prevalence of the habit of stretching and the moment in which they performed the stretching, but both variables were similar in both sexes.

The authors have added the identification of these non-significant results in the Results section.

Once again, thank you very much for the time spent and the interest shown in this work; as well as in the positive evaluations you have given of it.

Receive a warm greeting,

The authors.

Round 2

Reviewer 2 Report

The revised manuscript does not described sufficiently.  #1 question which is the main point of this manuscript may not be overcome at this stage. If not, the results of the study are not credential as scientific article. 

#2 is not the sample size issue. The method should describe the reasonable assumption that the results of selected subjects would be same as those of unselected ones.  Otherwise, if the subjected had been selected more or less, the result would be different. 

Author Response

Dear Editor and Reviewer of Journal of Clinical Medicine:

Thank you very much for your suggestions and contributions to improve the quality of the manuscript. Following your indications, we respond, point by point, to the reviewers' comments.

In the text, all the modified or added sentences have been written in red to facilitate the correction by the reviewers.

  1. The revised manuscript does not described sufficiently. #1 question which is the main point of the manuscript may not be overcome at this stage. If not, the results of the study are not credential as sicentific article.

The authors have rewritten the limitations of the research, objectively transmitting the representativeness of the results obtained. Despite having made this modification, the authors are not sure if we have fully understood their suggestions for improvement, if the changes made are not adequate, could you be more specific with your demands? 

  1. #2 is not the sample size issue. The method should describe the reasonable assumption that the results of selected subjects would be same as those of unselected ones.  Otherwise, if the subjected had been selected more or less, the result would be different. 
The authors have completed the Material and Methods section with this detail. As in the previous point, the authors are not sure if we have fully understood your suggestion for improvement. If we have not done it correctly, could you be more specific with the modifications to be made, please?    

Once again, thank you very much for the time spent and the interest shown in this work; as well as in the positive evaluations you have given of it.

Receive a warm greeting,

The authors.